# BLESSING: Exploring the Religious, Anthropological and Ethical Meaning

**Roger Burggraeve**

Faculty of Theology and Religious Studies, KU Leuven, BE-3000 Leuven, Belgium; roger.burggraeve@kuleuven.be

**Abstract:** The point of departure for this essay, which reflects on the religious, anthropological and ethical meaning of the act of blessing, is the multifaceted tradition of all kinds of blessings in the Catholic faith community, both in a sacramental and non-sacramental context. To properly understand the act of blessing, it is necessary to outline the existential and religious background of the blessing as an experience and condition. Starting from the general biblical background of blessing as an earthly reality, attention is paid to the transition from the implicit to the explicit religious meaning of blessing as a gift. Subsequently, the act of blessing in its bi-dimensional modality, namely as word and gesture, receives the necessary attention. This is accomplished by a shift from a theological to a philosophical understanding; this is anthropological and existential understanding of blessing. First, the specificity of the blessing as a language event is examined. Then, the bodily and possibly material form of the act of blessing is explored phenomenologically. Thus, it will appear that what is specifically Christian also has universal significance, is literally "catholic", that is, "kat' holon", meaningful "for everyone". Last but not least, consideration is given to the "power" of the act of blessing, both its "founding" power and the risk of magical derailment.

**Keywords:** being blessed; act of blessing; speech act; performative language; expressivity; bodiliness; power of blessing; magical derailment; phenomenology; ethics

## 1. Multifaceted Tradition of Blessings

At a first glance, it becomes clear that the Catholic faith community knows the practice of all kinds of blessings. This is fully reflected in "De benedictionibus" (1984) (*Blessings*) of the post-conciliar revised "The Roman Ritual" (1984) (Weller 2017): from the sending of missionaries to the blessing of a table, a house, a tabernacle, a cemetery; the blessing of a mother before childbirth, the blessing of the sick and aged; the blessing of animals (horses, pets, etc.) or of tools. There are blessings for places, things (objects such as 'founding stone' or vehicles; buildings such as seminaries, new homes, factories, facilities for care and relief, etc.); individuals, and groups or communities; and for 'special circumstances.' There are also blessings for objects or things that are erected or used in places of worship (churches), such as a baptismal font, as well as for objects of popular devotion, such as scapulars and neck crosses. We can distinguish between blessings in the context of parish life, blessings in family life or religious communities, blessings in the context of public life (for example, of a fire brigade), and blessings for general use. It is clear that blessings exist not only within but also outside the sacramental and official liturgical context (Davison 2014).

Many ecclesiastically recognized blessings fall under the heading of "sacramental". "These [sacramentals] are sacred signs which bear a resemblance to the sacraments. They signify effects, particularly of a spiritual nature, which are obtained through the Church's intercession. By them men are disposed to receive the chief effect of the sacraments, and various occasions in life are rendered holy" (Vatican II, *Constitution on the Sacred Liturgy* 1963, n° 60) (Flannery 2014). They are instituted to sanctify ecclesiastical offices, states of life, and all kinds of circumstances of Christian life. It is important to point to these sacramentals as an opportunity. In addition to the blessings that belong to the recognized

sacramentals, all kinds of blessings that accompany daily life have developed and continue to develop in popular religion. There are also circuits of alternative blessings, for example, in the Christian women's movement, which believes that traditional blessings affirm the subordination of women too much (Walton 1985). It is not the purpose of this essay to go into this diversity and its tensions. Rather, our approach tends to be global and focuses on the act of the blessing, whatever its form, the minister or "performer", and the recipient.

## 2. Blessing as Being Blessed: Earthly and Religious Experience

To properly understand the Christian act of blessing, we need to look at how blessing is taught or evoked in the Bible. We mainly focus on the general biblical interpretation, without offering a detailed exegetical study of the different modalities and contexts of the blessing (see: Mitchell 1987). It is striking how, in the Scripture, there is never talk of blessing as an act (performance or ritual) unless the idea of blessing as a state and gift experienced by a human being is supposed.

### 2.1. An Earthly Reality That Is Experienced as a Blessing

Starting from the Old Testament, blessing can be described as a positive tangible reality: health, water, life, wealth, fertility, wholeness, and well-being, in short, everything that is good and beneficial. Synthetically, the Bible makes a direct connection between blessing and life, in the sense that blessing stands for the fullness of life as an earthly created reality.

Even today, people still use the word "blessing" to denote the good that comes to them in some way. Positive things such as (sufficiently) good health, prosperity, children, a successful operation, a caring partner, and loyal grandchildren are experienced as a blessing and happiness, as "gifts" that happen to us. In our striving for self-determination, these experiences bring us into contact with the heteronomous, the coming from elsewhere, that which is beyond our control. There is something of a "fortunate fate" in this, which happens to us despite our power and is therefore undeserved—in contrast to the "unfortunate fate", which is all about tragedy and curse. Not only the special but also the ordinary "things" that make us happy are both of the order of the gift and of the order of abundance. That is precisely why people continue to call those things a blessing in a profane context.

### 2.2. From Implicit to Explicit Religious Significance of Blessing

The question is, how are blessings and God connected? At first glance, the mentioned forms of being blessed seem to have little religious significance. They concern earthly realities that people experience or interpret as blessings. It could be questioned whether a completely non-religious use of the word blessing is possible. After all, there is, at least implicitly—and in our secular world—a reference to a "power" that brings good things to us. People recognize that one is not the almighty, nor the initiator, nor the alpha and omega that determine meaning and purpose. Blessing is being blessed. This passivity means that people almost naturally interpret the word "blessing" religiously, or at least presuppose the religious dimension. In everyday language, a "coincidence", as "good luck", is regularly connected with the expression "from-god-knows-where . . . ".

The same implicit religiosity also applies to the "pronounced blessing", the act of blessing. In blessing, both for those who promise it and for those who receive it, there is always a reference to a heteronomous element: something that comes from elsewhere through something or someone else. Therefore, the word "blessing", and certainly the act of blessing, is almost naturally interpreted as religious—or at least the religious dimension is assumed. After all, the one who blesses appeals to the unconditional and absolute, without being able to find it in oneself. By blessing one refers—in the blessing itself—as it were, to an absolute reliability, which far exceeds the one who blesses. In the blessing, one also connects the person on whom the blessing is pronounced with "the absolute". One cannot pronounce a blessing unless one does not also, at least implicitly, resort to a "higher" authority that guarantees the outcome of the blessing. In other words, by blessing,

the person who blesses appeals to his explicit or implicit conception of God. In this respect, we discover how, in the blessing, a theological meaning is implied, that is, a reference to the divine, although the one performing the blessing and the one receiving the blessing are not necessarily aware of this theological significance.

The biblical blessing implies an explicitly religious meaning: blessing is always "from God". As creatures created by a Creator, we experience creation as a gift given to us "from elsewhere". Our creation is not a diabolical curse but a divine blessing. This implies that the explicit reference to God-Creator is a qualified reference, namely an experience of God as benevolent beneficence. God is not a neutral explanatory principle, but an ethically qualified reality. The earthly fullness of being and life is simultaneously seen as a sign of divine grace: God's ethical goodness is a blessing to us! In other words, this implies that not just any idea of God can be associated with blessing. This is only possible if God is seen and experienced as a graceful God: a God who is a blessing himself, and of which an expression and 'radiation' can then be found in creation.

With Matthew Fox, we can also call creation the "original blessing", in contrast to the "original sin" (Fox 1983). Based on today's sensitivity to the environment and earth, he argues for a paradigm shift. While, in the past, too one-sided an emphasis has been placed on the Fall or Original Sin and the curse associated with it, the gift of creation deserves at least as much attention as the paradigm of sin and redemption. While the fall and redemption strongly emphasize the negativity of failure and human smallness, a swing to the creation paradigm allows us to place a stronger emphasis on power and creativity. Humans were created as connected to the world, and the world was given to them as an environment in which it is good to live. Creation is the blessing of God Himself for humans, and moreover, it itself is also blessed by God. Insofar as creation is God's blessing, we can also experience the world religiously as a gift: "God is the giver of all good". In our creation, we relate to the world in gratitude, grateful as we are for the gift received. In addition, through the gift of creation, the human being also shares in the divine energy deposited in creation. Creation is a divine blessing to humans because it also bestows divine energy and creativity on them, as God also blesses living beings with dynamism and fruitfulness (Gen 1:22.28). God blesses humans through creation, making the world and the stream of life a blessing and gift to humans. The redeeming and liberating grace of God, which Christians especially receive and experience in Christ, must not be separated from the way in which God gives himself to humans in creation. Based on our belief in God's creation, blessing is the active, creative presence of God in his creation and his creatures: in nature, in all living beings (plants and animals), in fellow human beings, in our own lives and in its joys (without this leading to a deification of creation). Without becoming blind to the dark side and the ambiguity that is also attached to creation—as a finite reality—which means that it is sometimes experienced as chaos and curse, creation is, for those who profess that God is the Creator of heaven and earth, an essential act of faith to receive and experience the gifts of creation as blessings from God. We may enjoy creation and life, because they have been given to us as a blessing from God! That creation blessing is not over with the Fall. Even after the flood, in which the very existence of the earth was threatened, the divine blessing of fertility is confirmed. Moreover, throughout the Noah story, the creation blessing evolves into a covenant blessing, because God makes a covenant with Noah and his descendants, and with all living beings (Gen 9:11).

In the Gospel, the New Testament, the idea of God's grace as blessing, is consistently included and extended in its own way. Jesus also reveals God to be a "blessing full of grace". According to the synoptic Gospels, this is evident from his proclamation of God's kingship ("Basileia tou Theou") (Mc 1:15) (Merklein 1981, pp. 17–45). This implies that Jesus never speaks of God per se as some sort of neutral ontological given or indifferent "fact of being", although this is the grandest and most powerful fact of Being. He always speaks of God in a specific way, namely by always connecting God with the idea of kingdom, or rather with "reign", for he does not mean a place but an active event. He then assigns a paradoxical meaning to this active "reigning" by, as it were, turning it inside out and connecting it with

a serving and liberating approach to people. Through this "reversal" of the worldly power category of "dominion", Jesus proclaims a near God who empties oneself of one's majesty that causes "fear and trembling" to associate with the "poor, weeping, hungry, crushed, persecuted" (cf. Beatitudes') (Mt 5:1–11). They are bestowed with gifts by God Himself, comforted, satisfied, raised up, restored: "for theirs is the kingdom of heaven" (Mt 5:10b). This echoes the way in which God revealed Himself as the Merciful since Abraham in the First Testament. This mercy is to be understood as "uterinity" or "wombness", in the sense that "rahamim" (mercy) goes back to the root word "rehem" (uterus, womb). In the words of Emmanuel Levinas: "What is the meaning of the word Merciful (*Rahamim*)? It means that the Eternal One is defined by Mercy. *Rahamim* goes back to the word 'rehem', which means uterus. *Rahamim* is the relation of the womb to the other, whose gestation takes place in it [to be born] ["trembling of the womb where the other is in gestation in the same"] (Levinas 1994, p. 142). *Rahamim* is maternity itself. God is merciful; God is defined by maternity. Perhaps maternity is sensibility itself [i.e., touchability and vulnerability by and for the other], of which so much ill is said among the Nietzscheans" (Levinas 1990, p. 183). By his proclamation of the Kingdom of God, Jesus not only proclaimed God's mercy, but also put it into practice and incarnated it in his whole being. Jesus 'is' what he does and says: "agere sequitur esse" becomes "esse sequitur agree", being follows from doing and saying. Not only is Jesus the face of God's *Rahamim*, but through his humanity he makes divine mercy tangible and sensible in his flesh and blood in this world, among people. He literally "does" God, namely through all kinds of acts of recognition and appreciation, communion at the table, exorcism of all kinds of "evil demons" who occupy and obsess humans, forgiveness of sins, healings and raising people from the dead ... Through his "ethics"—being merciful to vulnerable people—Jesus reveals and embodies God's *rahamim*. Jesus' ethics is our grace. Therefore, it becomes clear how God's ethical quality—the One's "extravagant merciful love"—is our blessing: Jesus Christ is God's incarnated blessing for us!

## 3. The Act of Blessing as Language Event

It is this proclaimed divine blessing of creation and life, and of the "Reign of Love", that brings believers to the act of blessing. Henceforth, we turn our attention to this religious act of blessing as a human act. It is important to note that we do not primarily aim to use a Christian–theological approach (Davison 2014), even if that approach is not absent (Greiner), but rather pursue a philosophical, i.e., phenomenological, anthropological and existential understanding (Austin, Dolto, Ginters, Ladrière, Marcel, Searle). First, we focus on the blessing as a language event and then, in a subsequent part, we pay the necessary attention to the "incarnation" of the blessing through gesture and its material modality, namely the material elements that possibly accompany the word and gesture of blessing.

### 3.1. Blessing as an Elementary Language Act

With Dorothea Greiner, we can say that blessing as a language event is an "elementary language act" according to distinct aspects (Greiner 1998, pp. 36–38).

First, blessing is elementary because it is a simple act. According to the Latin term, the blessing does nothing but say something good: "bene-dicere". Three words suffice for the religious blessing: "God bless you". This formula is recognizable in many religious traditions and contexts. In addition, the blessing formula is usually accompanied by a simple gesture, sign (cross) or concise ritual, without frills and mannerisms, and is short and powerful.

Next, the blessing is an elementary act because it relates to a basic human desire, namely the pursuit of acceptance and healing. People are looking for places, persons and figures of ultimate trust, of ultimate surrender to lasting meaning and future, to life stronger than death, to a love stronger than all disaster and evil. As a language event, the blessing creates space for this ultimate trust, the belief that one is ultimately safe and saved, that life is worth living in the end. The blessing uniquely meets our great desire—our

hunger and craving—for fullness of life. In this regard, there is a deep connection between blessing and promise. The promise transcends the ruptures that certain experiences create in our existence: negative ruptures of evil that seem irrevocable. The promise leaves open the possibility that the irrevocable can be revoked, that one can trust in the future in spite of everything. Hence, the Christian faith professes God as a promise of healing, redemption and reconciliation through Christ, through which the unexpected can be born and everything can become new. This is the ultimate, far-reaching meaning of the religious Christian blessing.

Thirdly, blessing is an elementary act because, as a language act, it has a creative and "founding" meaning. Blessing does not only imply proclamation, but is also the commitment and realization of its own promise. What the blessing pronounces is also accomplished through the blessing itself. Through the blessing that is spoken to persons, they experience that they are or become blessed. In this respect, the blessing is a "performative" speech act (Searle 1979). It is a "language act", with the emphasis on "act", because the blessing accomplishes something. Moreover, it concerns a "performative" act, to be distinguished from a descriptive or informative act. An informative statement simply reflects a certain state of affairs, communicates a certain content of knowledge, and affirms that a certain "given" (object to person) behaves in a certain way, that something occurred in some way, and so on (Austin 1962). A performative statement affects, or at least intends to affect, something in the person to whom it is spoken, and also in the person who utters it. The language act itself creates a new situation that was not there before. Through what is said, both the addressed and the speaking person are involved in the new situation. As a result of this, they change, literally become "different" and are also called on to react and to "do" something with what is said (Ladrière 1973). From this distinction between informative and performative language, it is clear that the blessing is a performative speech act. The word of blessing is an act that effects what it communicates, namely that the person to whom or about whom the blessing is pronounced is the beneficiary of God's love. That the One first loved us (1 Jn 4:19) is not only the core of the Christian message, but also the ground and transformative effectiveness of every Christian blessing. As a speech act, the religious (Christian) blessing introduces the receiving person to the "experience" that one's existence has ground under its feet, that one's existence is anchored and surrounded by a Reality to which one not only can give oneself, but from whom one can also gratefully receive and cherish oneself.

### 3.2. Specificity of Blessing as a Modality of Religious Language

After this exploration of blessing as a speech act, we now want to examine the religious specificity of blessing by distinguishing it from wish and intercession (Greiner 1998, pp. 43–54). At first glance, the blessing seems closely related to the wish, in the sense that whoever blesses someone wishes the other well. Sometimes, a wish is even called a blessing. Both the wish and the blessing are marked by desire; they are a form of eager anticipation and, therefore, of hope. Typical of the wish is that it relates to the relationship between two people: I wish you all the best. The wish is a dual reciprocal relationship: "I—you". In the wish, the emphasis is on the other (as object) and on the I (subject) that expresses the wish. Even if the I is omitted, as in "Good luck with your marriage!", the expression means: "I wish you good luck with your marriage!" The wish does not require a third party to act as guarantor. The wish leaves open who or which body is responsible for the addressed "happiness": fate, God, the addressee her- or himself, etc. Just think of the Happy New Year's wish: "I wish you a very good new year". Remarkably, those who wish for a happy year do not guarantee it themselves. The guarantor is left undetermined. The religious blessing, however, manifests itself unequivocally as a "triad" or triangular relationship: it involves a relationship between three persons. In that blessing, there is always a clear reference to a third person, besides you and me, namely God (or his "representative"). The emphasis is not on the I, the subject who blesses another, but on the One who guarantees the blessing: "God grant you a good year"; "God grant you that your marriage succeeds".

In this respect, the ego withdraws to make room for God: a form of human anachoresis or kenosis. Moreover, God cannot be equated with fate ("fatum"). It is not just "something or someone in general", or "fate" as "fortuna", that is invoked. After all, by fate, the object of the blessing is anything but assured. Fate can give all or nothing, be lenient or brutal; fate is, by definition, indifferent, neutral, un-preferential, wild and unreasonable, and therefore full of tragedy ("doom"). In the religious blessing, it is always a loving God who intends true good for human beings. We know this from the Biblical proclamation of the good news. However, the blessing is more than proclamation, for it also declares—and by pronouncing, also accomplishes (as stated above)—that God as "embracing grace" wills the very best for the blessed person (community, group, etc.).

This shows how the blessing is, par excellence, a religious act. The speech act of blessing introduces God, as a third person as essential, in this speech. It is not the one who pronounces the blessing that bestows the blessing. God is the founding referent of the blessing. The one who pronounces the blessing does not refer to her- or himself, but to God, who is the real subject—the real Agent—of the blessing. The blessing is primarily concerned with the relationship between God and the blessed. God and the human being are brought into a personal relationship with each other, which is true religion. This connection is not established by the blessed person (or group) but by the blessing pronounced by a person other than the blessed person(s). At the same time, the person who blesses withdraws as much as possible, because that person is only a "mediator": only God is the source of the blessing. This is also shown in the famous Aaron's blessing: "The LORD bless you and keep you; the LORD make his face to shine upon you, and be gracious to you; the LORD lift up his countenance upon you, and give you peace" (Num 6:24–26). With this, we encounter a double "extra nos:" the blessing brings people "outside themselves", both the blessing one and the blessed person. The one who blesses does not bless oneself, just as the blessed does not bless oneself. Whoever blesses points away from oneself in a twofold way: to the blessed other and the Blessing One. The blessed person is not the source of the spoken blessing; he or she is blessed by another. This is precisely the event of grace that breaks open humans from their self-confinement and directs them toward an other than themselves: the divine Other. The person who is blessed is connected with the divine Other One, and so, too, with the human person who speaks the blessing. In this respect, the blessing is completely at odds with the autonomistic self-redemption tendency, which one believes one can possibly achieve through all kinds of techniques and methods (of "mindfulness" and others).

The distinction between wish and blessing has not yet revealed the whole uniqueness of blessing as a language event. However, related to the wish, the blessing is more than wish, it is also, and above all, prayer. Hence, another comparison arises, with intercession or supplication. This relationship is so striking that one does not always perceive the difference between the two. The gesture that often accompanies both shows the difference. When people pray with their arms open and upraised, we know that they are addressing God "in heaven", the true addressee of prayer. In intercessory prayer, people turn to God asking for help and assistance from someone, an individual or group. When someone blesses another person, for example a parent blesses their child, we see a different body movement. People who bless address the receiving person. They extend their hands directly to, or rather over, the addressee, or they lay hands on the other. Even the prayer for blessing a person differs from a direct blessing. In such a prayer for blessing, one does not extend the open arms and hands to the one who is the object of the blessing, but to God, to heaven, for the Holy and no one else is the addressee (second person): "Lord, we implore You: bless N.N. with your gifts". The mere blessing, however, refers to God in the third person and directly addresses the person (to be blessed): "The Lord (3rd person) bless you and keep you (2nd person)" (cf. supra). As said, this finds expression in the bodily gesture that accompanies the word of blessing: the language of the arms and hands gives shape to the language of the words!

*3.3. Bodily and Material Dimension of Blessing*

Through the comparison with the wish and supplication, we involuntarily arrived at the bodily form of blessing. Sometimes, the blessing is just a word, like Jacob's blessing of all his sons (Gn 49:28). Usually, however, the blessing, even if it comprises a strong word, is accompanied by a gesture or an action in the form of a well-defined sign, such as the laying on of hands, the sign of the cross, or the sprinkling of holy water. Sometimes, but rather exceptionally, the blessing is performed only as a gesture, a bodily touch or an action, without any word. For the most part, word and gesture go together in an indissoluble unity. Moreover, material elements, such as water and oil, are often used in the performance of the blessing. There is no Christian rite where people come so close to each other as blessing. For example, in individual blessings, the hand of the person blessing touches the other person's hair and scalp. If one blesses by drawing someone with the cross, one draws this cross on the forehead (or on the hands and feet). Pouring the water over the head of the person being baptized is not as intimate as blessing that person with a bodily touch. In baptism, the material element of water stands as a mediating element between the person who is baptizing and the person being baptized. On the other hand, there is also blessing during baptism: the person being baptized is signed with the cross and there is the laying on of hands, anointing with the chrism. In the Effeta prayer ("Open up"), the various senses of the baptized are touched and anointed. In the sacrament of the sick, the sick are also anointed and touched. This means that the blessing is often not only gesture but also touch. Even if we have become accustomed to it, it remains a curious phenomenon. Usually, people try to avoid contact with strangers or those with whom they do not have a connection. In ordinary life, people will not touch someone's head or hair unless there is a more intimate relationship with that person. In the blessing, people do touch people even though they have never met them before. If we were to touch someone in that way in everyday life, we would apologize for it, and run the risk of being suspected or even accused of (sexually) transgressive behavior. In other words, the blessing is a ritual "sanctuary". The blessing creates a situation that allows people to come close to and touch each other without embarrassment or apology. In addition, the rite of blessing allows the bodily proximity to become religiously transparent. The purpose of blessing as a rite is that God comes close. After all, people do not say, while touching: "I bless you", but "God bless you" or "The Lord be with you". The one who blesses does not primarily express his own closeness through the gesture and the touch, but through divine closeness and touch.

This puts us on the trail of the expressive nature of the blessing. Gestures, signs and material ingredients embody the word of the blessing, making the content of the blessing tangible. In order to properly understand this expressive character, a reflection on 'expression' is first necessary. Before the so-called "linguistic turn" ([Rorty 1967](#)), the expression was often understood too spiritualistically, starting from the consciousness that already possesses an idea, essence or content in itself, and then expresses it in a certain form at a later stage. Such an approach implies that, on closer inspection, the form is incidental and certainly does not co-determine the preceding essence or thought. Under the influence of Descartes, the body was reduced to a "res extensa": the object of consciousness ("cogito"), source of the autonomous subject. As a result, the body could also be regarded as an instrument that one can control and which one can use to realize one's "projects of meaning". Then, the thinking consciousness becomes the active "signifier" and the body the object and instrument of signifying. Since the linguistic turn in philosophy, we have gradually started to think differently about this, although the turnaround has still not fully taken effect. A careful study of speech makes us understand that the language in which an idea expresses itself also helps to create the idea. The form is not only a decoration, but also a "founding" for the content. Our body is not incidental but essential and constitutive of our humanity. It is not only an object of "signification", it is itself a "signifier". Even if we cannot reduce our humanity to our bodiliness, which would create a flat materialism, we do not only "have" a body, but we also "are" our body (Gabriel Marcel) ([Troisfontaines 1968](#), pp. 173–75). The body provides us with a number of meanings that

invite us to look at things differently. This is, perhaps, the truly expressive character of the human bodily experience. Our body as a "lived body" also characterizes our intentional giving of meaning. In itself, the spirit is an abstraction; it only exists thanks to the living and lived body. The human body has a spiritual dimension in the sense that it facilitates and orients the spiritual—and thus is also a source of meaning.

When applied to the act of blessing, this means that the bodily form not only gives shape to the content, but also co-establishes and creates that content itself. This means that blessing is a special kind of act, namely an "expressive act" ("Ausdruckshandlung") (Ginters 1976, pp. 11–18, 36–44). This action must be distinguished from instrumental action, which is based on a functional relationship between the goal, namely the effect one wants to achieve, and the means to achieve this goal, namely the action. The primary goal here is an outcome that lies outside the action itself. In addition, the cost–benefit analysis plays a central role. The value criterion lies in the balance between advantages and disadvantages, or rather the positive outcomes must prevail over the negative outcomes. However, our actions cannot be reduced to instrumental functionality and pragmatic effectiveness. They are much richer and multidimensional. After all, in addition to instrumental acting, there is also "intrinsically meaningful acting". Such actions are performed for the sake of the value or meaning of the act itself and not for some beneficial objective or external effect. The desired effect lies in the action itself: means and goal coincide. In this respect, the value of the action lies in the quality of the action itself. We can think, for example, of play and of artistic expression in any form. Moreover, we perform many instrumental actions not only because of their usefulness, but also because of the intrinsic, qualitative sense of the action itself. Then, those actions become "sur-determined" in the sense that they acquire an "additional meaning" that elevates them above everyday neediness and usefulness. It is remarkable that they then also acquire their expressive character from the reversal of the utilitarian benefit–cost ratio, in the sense that they often cost more than they produce in benefits. That is why outsiders often call them a form of waste: "costly expressive acts" ("kostspielige Ausdruckhandlungen"), which are not only "expensive", but often also imply a form of exuberance and exaggeration, which can go as far as to be wasteful (Ginters 1982, pp. 94–97). Even if they are not wasteful, they are usually economically unnecessary and even useless. Last but not least, the expressive nature of the expressive actions also has a relational dimension, in the sense that they direct what is expressed to another, and thus not only enable but also realize relationships (cf. infra).

The act of blessing is pre-eminently an intrinsically meaningful act that is expressive in word and gesture. By audibly and tangibly shaping the divine blessing, the act of blessing itself realizes that blessing "here and now". In this respect, the blessing is not an "extrinsic" but an "intrinsic sign". Both types of signs refer to something other than themselves. With the extrinsic sign, the "other reality" lies outside the sign itself, as becomes clear in all sorts of functional, conventional signs in society. In the intrinsic sign, the signified reality coincides with the sign itself, so that the signified—the reality of blessing—is fulfilled by the act of blessing. The gesture of the blessing, together with the audibly pronounced word and any material elements, can therefore be labeled—by analogy with the sacraments—as an "efficacious sign" ("signum efficax"): it realizes what it signifies. To bless someone by the laying on of hands means that the blessing is realized here and now by the sign on the person in question, so that the person "is" also a blessed person from the moment of the blessing. Moreover, the act of blessing as an audible and tangible expression of God's blessing connects the blessed with the Holy One as well as with the one who performs the act of blessing, and with the community in which the blessing may be performed.

## 4. Power and Powerlessness of the Blessing

The expressive character of the act of blessing leads us directly to the "power" of the benediction, which deserves separate and explicit attention (although it already appears in the above reflections). On the one hand, we reflect on the special power of the blessing; on the other hand, we examine the risk involved.

### 4.1. The "Founding" Power of Blessing

First, there is the creative power of the blessing. In contrast to the curse, the French psychoanalyst Françoise Dolto (Pohier 1985) puts us on the trail of the "founding" power of blessing. Through her fellow psychoanalyst Chertok, who specialized in suggestion under hypnosis, she came to understand that the curse is something against which hypnosis is powerless. The curse forms such a harsh reality that even hypnosis cannot get through or overcome it. Someone who was cursed at conception cannot be helped by hypnotherapy. A curse, which the child knows about through its parents and applies to the father of the child and his offspring or to the child himself, exerts an indelible influence on the whole of his life, on the fruits of his labor and his sexuality.

Conversely, this also applies to the act of blessing and the being blessed that forms the basis of this. In order to become human, one depends on the blessing that others give us. One is dependent on the blessing of others in order to experience one's own life as a blessing. As the curse kills, so the blessing creates life. Even if a blessing does not immediately help, it is, for the blessed one, the assurance of protection and hold in trouble, a promise for the future. The blessing is not an incidental thing, such as clothing, it is so strongly intertwined with existence that it determines the positive or negative sense of what a person is and produces. You can meet people who have a radiant feeling that everything will work out. Even though they may not have much reason to believe this at present, they firmly believe that the future will be positive, "because someone told me so, and I trust it!" Sometimes, you hear people say: "I will succeed because my father (or my mother) said that I will succeed. I may not experience it anymore, but my children will certainly experience it". However, this feeling of confidence is not the result of one's own performance. It is not one's own creation but a gift from someone else: a blessing given to me. This is the ground for the deep emotion of knowing oneself to be blessed, despite all that is happening!

The Bible also assumes the efficacy and power of the blessing. Four aspects stand out regarding the impact of the blessing on the blessed (Greiner 1998, pp. 143–48).

First, there is the strengthening effect, which consists of strengthening the blessed. Thanks to the blessing, people, individually or collectively—for example, the people of Israel—find resilience and become strong and fruitful. Sometimes a new name, and thus a new identity, is the sign and confirmation of this. For example, after wrestling with the stranger (an angel of God), Jacob asks the angel for his blessing (Gen 32:23–33). However, before the angel blesses Jacob, he also gives him a different name, and thus a new identity and meaning, namely "Israel": he who wrestles with God and was victorious. The blessing that follows confirms the new identity and gives strength to face the new existence and the new mission.

Second, there is the protective effect, in the sense that the blessing offers shelter against all kinds of evil, threat and suffering. The blessing protects against misfortune and disaster, with the risk that the blessing will be perverted by the recipient into a magical power that one tries to exorcise or bend to one's own will (more on that later). The blessing stems from the acute awareness that reality and history, both great and small, are not always rosy, but are often marked by obstinacy, boundaries and setbacks. In a perfect world, humans would be omnipotent, omniscient, and omnipresent, without question and uncertainty, so that they would not have to step outside themselves and knock on someone else's door. Whoever asks or receives a blessing confesses that he is not the "master and possessor of this world". The blessing presupposes humble and small people, not masochistic people who indulge in their smallness and wallow in it narcissistically, but realistic people who realize not only that they are finite and fragile but also that they cannot survive on their own. Asking for a blessing is a sign of realism and humble self-knowledge without self-abasement!

Thirdly, there is the healing effect: the blessing offers redemption. After all, people are also personally afflicted by all kinds of calamity and evil, especially by their sinfulness and weakness as ethical beings. Today, emphasis is often placed on finiteness and thus on error and failure. However, humans, as ethical and therefore free and responsible beings,

are also beings of evil and immoral actions. Humans are not innocent creatures; they are "sinners": "fallible beings" with a wounded freedom (Ricoeur 1986). This implies that they need redemption and healing from their sins and freedom. That, too, belongs to the intent and power of the blessing: to heal people from sin and evil, to deliver people from the guilt that traumatizes them.

Last but not least, there is the community-building effect of the act of blessing. Thus, in the Bible, the blessing is the beginning of the history of the covenant between God and Abram (and Israel) (Gen 12:2–3). A blessing always shows a binding force, as indicated above. It establishes involvement and communication between the "partners". Thanks to the one who addresses the blessing from God about (to) the addressee, involvement also arises, both between God and the addressee and between God and the blessing person. However, involvement grows also between the blessing person and the recipient, implying that, as a religious act, the blessing also establishes interpersonal interaction and relationality. In other words, the blessing not only contains a statement and declaration of a promising and merciful perspective, it is also a relationship event that makes clear: "You are not alone. We won't let you down, we love you!" Not only is there the divine Other, and compassion for people, blessings also connect the one who utters the words of blessing with those people. Blessing someone is impossible, or at least lying, if one does not have a positive attitude towards the other person and does not want the best for him or her. Hence, the curse is radically opposed to the blessing. After all, a curse not only pronounces a curse on someone else, that curse also expresses the aversion and rejection of the one who pronounces the curse. In the blessing, I not only wish the other person the best, but I also express that I am close to the other and that I support the blessed person.

### 4.2. Risk of Magical Derailment

There is also a downside to the effectiveness of the blessing. After all, it can derail into magical claim and even obsession. Because of their finitude, people tend to use the blessing as a magic spell, the mere utterance or muttering of which they believe can bring about its intended good. Blessing formulas are used to influence and control certain supernatural or extra-natural forces. Some even claim that blessings always function as spells, and therefore they should be rejected. This extreme view, however, confuses blessings with "incantations", which turn wishes into demands by expressing them in imperative form. The incantation presupposes a divine or demonic power to which one has no direct access and which one, therefore, cannot easily understand nor immediately control. By means of a kind of "magic formula or ritual" people try to get a grip on that power. One hopes to be able to force that power to please the person who uses the magic formula, or at least not to harm that person (which is certainly true when this concerns an "evil power").

However, this must be distinguished from authentic dealing with blessings. After all, authentic religiosity is not based on fear and therefore does not resort to attempts to "coerce" divine powers. Magic seduces or even bribes the "gods" into action, while authentic faith rests on trust and hope without certainty. The blessing remains a prayer and never becomes a "demand", in the sense that one entrusts oneself in surrender to God's love. Therefore, authentic blessing requires a culture of impotence, strange as this may sound. Only if, in pronouncing or receiving the blessing, one accepts that one cannot exercise power over the result of the blessing, can one avoid getting bogged down in the despair of the convulsive grip on the divine. Then, people accept that the blessing does not work immediately, but sometimes only much later. Then, people also accept that it may work completely differently than expected, or that it might not even work. As Job confidently puts it: "The LORD gave, and the LORD has taken away; blessed be the name of the LORD" (Job 1:21b).

A special aspect that can protect us from the magical derailment of the blessing is the ethical appeal contained in the blessing. The blessing of the divine fullness of grace, to which the blessing refers both in its origin and in its purpose, immediately implies an invitation to the person pronouncing the blessing to act blissfully. In fact, this is about

more than the desire that spontaneously bubbles up from being blessed. This is not about a possibility but about an incentive, even a "commandment": whoever receives a blessing from God should not keep it to oneself, but must pass it on to others, not only by pronouncing blessings on others but also, and especially, by treating others benevolently, namely by treating them with incarnated "deeds" of blessing.

### 5. To Conclude: Blessed to Bless

"No one knows the power of the body" (Atlan 1996, p. 209). Paraphrasing this statement by Spinoza, we dare to put the following into words at the end of our quest: "No one knows the power of blessing!" This can be expressed, both humanly, religiously and Christianly, in word and gesture! That the religious act of blessing, right down to its anthropological basis as an event of language and body, has a unique meaning and special effect, became phenomenologically clear in a multifaceted way. This phenomenological analysis also makes the religious blessing accessible and valuable "for everyone"; it is literally "catholic"—"kat' holon", with a universal meaning. From the general biblical perspective, it also became clear that the experience of religion in general and the practice of the Christian faith in particular needs the "blessing of God" in all its facets: the gift of life and creation, love and redemption. Moreover, this "grace" needs moments of condensation through which God's blessing becomes concrete and tangible for earthly beings. Blessings are such condensing moments. Through word and sign (and matter), they embody God's proximity. Moreover, that 'blessing from God' not only invites surrender but also makes that surrender possible. Precisely because of this, the blessing also provides the strength needed to get up and carry on, and even more so to take on the ethical commitment of blessing others. A synthesis of grace and ethics occurs: "You are blessed to bless; not only in word and gesture, but also and above all in deeds!"

**Funding:** This research received no external funding.

**Institutional Review Board Statement:** Not applicable.

**Informed Consent Statement:** Not applicable.

**Data Availability Statement:** Not applicable.

**Conflicts of Interest:** The author declares no conflict of interest.

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
