# Peer review of "BLESSING: Exploring the Religious, Anthropological and Ethical Meaning"

_religions, doi:10.3390/rel14050599_

Round 1

Reviewer 1 Report

This is generally a helpful account of the theology and practice of blessing.

I would have liked you to state what contribution you thought you were making to the literature on the topic - what are you saying that has not been said before, or if it has been said before needs reiterating in face of others saying the contrary?

I noticed that you referred to nothing published within the last 25 years.  Blessing is a neglected topic in serious theological work but I think you either need to show awareness of what has been said more recently or at a minimum remark on the absence.

P2 line 10 - here and in several other places in the review copy there was an error in the formatting producing a gap in the middle of the line - it may be a new paragraph was intended

P5 - a performative statement does not have to effect something in the person to whom it is spoken.  A jury foreman uttering the world 'Guilty' indeed changes something in the world, but that is irrespective of whether the defendant hears it.

p7 - immediately after the reference to Rorty, "we often understood expression too 'modern'" does not make sense. Likewise later in the paragraph "Our body as a “lived body” also characterizes our intentional giving meaning."  You may also want to think whether any further changes to the paragraph would help the reader.

P8 - I found the paragraph on cursing unpersuasive.  For example, it was not clear whether you believed the effect of a curse depended on the person cursed knowing about it.  In the case of blessing an infant, is that only effective through its effect on others or if the infant is told about it subsequently?  Would you say the same of baptism?

In the last sentence of the conclusion, "Synthesis of grace and ethics" lacks a verb

Reviewer 2 Report

The article’s strengths centred around the consideration of blessings from an anthropological perspective. This theoretical approach had merit, looking to understand the meaning of blessings in their various existential and material significances. The performative aspect is an important one to explore, and some interesting insights in this regard came out, especially in relation to the idea of blessings as language event. The article was well written and had a logical flow.

Nonetheless, there were significant areas of weakness, particularly in critical engagement with recent scholarship, which was distinctly lacking. The most recent reference in the bibliography is from 1998. Overall, much of the discussion took a theological tone, drawing on biblical references, but not engaging critically with key issues and recent academic research. For example, there was no interaction with scholarship on implicit religion in the section on explicit/implicit religious significance of blessings. Additionally, although there was some recognition of power issues and potential negative implications of the ritual of blessings in the later sections of the piece, no sustained attention was given to the impact of blessings in relation to universals in the sense of equality – e.g. how is gender performed in relation to blessings? If women are excluded from offering blessings, or are only recipients, what impact does this have in terms of gender and equality? Walton’s work is mentioned, but only cursorily in the article. Also, if same-sex unions are excluded from blessing, what are the implications for the ritual and its significance in wider social and ethical terms? The relationships between power and ritual in these regards was largely missing and did not engage with current debates and scholarship in these areas. There was also the suggestion that the physical touching element of a blessing could be considered sexually transgressive behaviour outside of the ritual context, which was less than helpful.

Reviewer 3 Report

Thank you very much for an interesting paper on the act of blessing in the Catholic tradition. This is an important field not only in terms of the Christian piety and living one's faith, but also within the scope of theological investigation.

The paper is based on the ground of Catholic theology. Besides the importance of the issue under examination, there are some crucial remarks explaining why this article should be deeply revised.

1. the language seems in many places to be spoken or popular, or even sermonic, but not scientific, making general statements unrelated to the evidence, e.g.:

p.6 - "It is not the one who pronounces the blessing that bestows the blessing. No, God is the founding referent of the blessing."

p.7 - "Usually we try to avoid contact with strangers or people with whom we have no connection."

p. 7 - "Our body itself is also spirit"

p. 8 - "Think, for example, of playing or artistic design in any form."

p. 9 - "We are not innocent beings, we are “sinners” (as the Hail Mary says)."

p. 10 - "Not only is there the divine Other who takes pity on you, but we, who utter a prayer of blessing, also feel connected to you."

The above shows the language is not clearly scientific, it more a parenesis than theological analysis.

2. In my opinion, the article lacks a good introduction and conclusion. Reading the abstract, I considered it a kind of introduction, but there is no such thing in the body of the article. The reader is not informed about the structure, hypothesis and methodology of further research.

3. The paper is very poorly referenced. In point 2.1 that is a part of Biblical background not a single biblical text is referred to. Point 2.2 contains some general views not grounded in evidence or necessary theological-biblical literature, e.g.:

p. 2. : "Even today, people still use the word ”blessing” - this is not Biblical background

p. 2 : "People recognize that one is not the almighty, nor the initiator nor the alpha and omega that determines meaning and purpose." - as above, it's a general view the author shares.

4. Some statements and phrases should be changed, explained and referenced, grounded in biblical-theological literature, e.g.:

p. 3: "Based on our belief in God’s creation, blessing is the active, creative presence of God in his creation and his creatures: in nature, in all living beings (plants and animals)" - how tih is seen by the author? is it pantheism?

p. 2-3: First and Second Testament:

Some propose that the Old Testament would be better referred to as the “First Testament” because old communicates a negative concept while first communicates a positive concept (cf. James A. Sanders, https://doi.org/10.1177/014610798701700202 ; John Goldingay (2003), Old Testament Theology, vol 1, p. 34). It does not seem to be a good idea. Even though the Old Testament books come first, the better name for this section of 39 books is Old Testament rather than First Testament. They detail the old law, the old way of gaining forgiveness, and the old way of receiving grace from God. The word old can mean “outdated and replaced,” which describes the old system that focused primarily upon the Jewish nation (2 Cor 13:14-15). The new system replaced the old with the sacrifice of Jesus. The new covenant (Jer 31:31 - new, not second!) offers salvation to “everyone who believes: first to the Jew, then to the Gentile” (Romans 1:16). We can fully appreciate the New Testament when we understand the Old. As St. Augustine's rule states it: Novum in Vetere latet et in Novo Vetus patet.

According to The Interpretation of the Bible in the Church, a document issued by the Pontifical Biblical Commission (1993) the adjective "old"  does not carry a negative connotation.

p. 4: "humane humanity" - what exactely does it mean? where doest it come from?

p. 4: at the end of point 2 the notion "salvation" and/or "redemtion" should also be appointed as intrinsicaly connected with the mission of Jesus

p. 5: - "In the religious blessing it is always about a reliable God who wants the best for the blessed and also guarantees it" - does the author intend to say that blessing guarantees happiness for the blessed person? any evidence for that conclusion?

p. 7: - baptism is shown as an example of a blessing, however in point 1 page 1 it was clearly said that blessings are not sacraments, but sacramentals - the crucial notions should be precisel defined and the author should avoid to mix them within the body of his paper.Moreover, on p. 1 Paul VI is mentioned as the author of the Constitution on the Sacred Liturgy - this is however the document of the 2nd Vatican Council, i.e.:

Second Vatican Council, "Dogmatic Constitution on the Church, Lumen gentium, 21 November, 1964," in Vatican Council II: The Conciliar and Post Conciliar Documents, ed. Austin Flannery (Collegeville, MN: Liturgical Press, 1975), sec. 14 (hereafter cited as LG).

Second Vatican Council, Constitution on the Sacred Liturgy Sacrosanctum concilium (4 December 1963),

Reviewer 4 Report

This is a well-written paper that follows a clear outline. It is well documented, with sources in French, German, Hebrew, semiotics, and speech acts.

It does not indicate, at the beginning, what it wants to accomplish. The title indicates the content: exploring the religious, anthropological and ethical meaning of blessings” but for what purpose?

It begins with the various kinds of blessings in the Catholic tradition, but this is well known, and Catholics do not need a lengthy scholarly explanation of their tradition–or maybe they do. Blessings are a practice (a minor one) not an important Christian belief.

Whatever its intent, this article is well done.

Round 2

Reviewer 2 Report

In the new amended version, the author has now included one source which is more recent than 1998, but as far as I am aware, it is a translation of Vatican II documents by Flannery, which were originally published as an edited collection in 1996. In this case, then, there is still no interaction with scholarly perspectives in the last quarter of a century. As the author says in their response, they view the article as ‘not an academic discussion with others’, but rather a ‘deepening’ approach. The author does not, in fact, use the term ‘deepening’ at any point in the revised article, and it is not clear what such an approach entails, as this is not set out. Normally, peer-reviewed journals do publish ‘academic discussions with others’ and writing that engages with other recent scholarship.

There are weaknesses in the clarity, structure, framing and use of evidence (both primary and secondary) in the article. As an example, in section 2 of the article, the author does not engage with standard reference works in biblical studies, and does not offer evidence in support of the assertion that ‘there is never talk of blessing as act (performance or ritual) unless the idea of blessing as state and gift, experienced by a human being is supposed’. Aside from the fact that this could be more clearly expressed, what texts are in mind, or support this position? In the entire section 2 on ‘biblical background’, only refers to 5 selected biblical texts: Genesis 1:22-28; Genesis 9:11; Mark 1:15 [NB ‘Mc’ is not a standard scholarly abbreviation for the Gospel of Mark; the term bless/blessing does not occur in Mark 1:15, only the Kingdom of God] Matthew 5.1-11 (and 10b). Considering the fact that there are more than 200 references to the Hebrew and Greek terms for bless/blessing in the Hebrew Bible and New Testament, how does this adequately provide a ‘general biblical background’? The discussion of Levinas in this section is the most interesting, but relates to the Hebrew term for mercy – rahamim – not blessing. As far as I am aware, Levinas makes no connection with the idea of blessing in the sections of his work referenced here. Even the author does not attempt to connect blessing to Levinas’ work on mercy, only glossing it at the very end of the section – ‘The One’s ‘extravagant merciful love’ – is our blessing: Jesus Christ is God’s incarnated blessing for us!’ This is tagged on with no meaningful point of connection.

Some points are also open to misinterpretation. In section 3.3, the author states:

Usually people try to avoid contact with strangers or those with whom they don’t have connection. In ordinary life people will not touch someone’s head or hair unless there is a more intimate relationship with that person. In the blessing people do touch people, even though they have never met them before. If we were to touch someone in that way in everyday life, we would apologize for it – also running the risk of being suspected of even accused of (sexually) transgressive behavior. In other words, the blessing is a ritual “sanctuary”.

Though multiple interpretations of this may indeed be possible, at least one interpretation could be something along the lines of: From an anthropological perspective, blessing allows touching to take place in a ritual context which would otherwise be considered inappropriate, and even sexually transgressive.

Who is the ‘we’ that is mentioned (‘if we were to touch someone)? If the ‘we’ is considered to be a priest, then the implication is that male priests in this ritual context engage in touching men, women and children under the category of ritual ‘sanctuary’.

Are these potential lines of interpretation ones that the author wants to suggest? Is the author aware of Marleen De Witte’s work on ‘touch’ in religion? De Witte M (2011) Touch. Material Religion 7(1): 148-155.

In light of the significant weaknesses of the article, and its own positioning as outside of academic discussion with others, it does not easily fit in a peer-reviewed journal, but may find a publishing home elsewhere.

Author Response

My article is conceived and elaborated as a "reflective essay", which thinks theologically but - from part 3 ("The act of blessing as language event") onward - mainly philosophically (anthropologically, phenomenologically, ethically) about blessing and the act of blessing. It does not want to have a discussion between authors, but - creatively using the insights of the basic work of Dorothea Greiner ("Segen aund Segnen") and of important basic authors as Austin, Dolto, Ginters, Ladrière, Marcel, Rorty, Searle... - a vision of blessing develop that is "universally communicable" and therefore gives everyone to thought. To clarify better this own "statute" of the essay presented, reformulations anbd minor additions were introduced in the Introduction (Abstract) and in the introduction to parts 2 and 3 of the essay. The keywords were also adjusted accordingly. Taking into account the reader's comments, for which thanks, some refferences have been also added. However, this does not change the nature of the article als a "reflective essay".

I leave it to the chief editor of the issue to judge whether my essay (in its third, final version) van be included in the isse of Religions.

Incidentally, I would like to note that the reader did not receive the second, improved version of my essay...

Reviewer 3 Report

p. 10: "We are not innocent beings, we are “sinners” (as the Hail Mary says)."  It is a theological construction that should be based on references to theological literature and not only limited to quoting the text of a prayer without any other justification.

All remarks from the 1st revision (in the pdf attached) were addressed by the author, however finally not intoduced into the body of the paper I got for 2nd reading. So I kindly ask to study once again the author's response attached and go on with the correction.

Author Response

The reference to "sinners" in Hail Mary has been replaced by a more general indication, referring to the work of Paul Ricoeur: "Fallible Man".